# Shape-Dependent Toxicity of Silver Nanoparticles on Freshwater Cnidarians

**DOI:** 10.3390/nano12183107

**Published:** 2022-09-07

**Authors:** Joelle Auclair, François Gagné

**Affiliations:** Aquatic Contaminants Research Division, Environment and Climate Change Canada, 105 McGill, Montreal, QC H2Y 2E7, Canada

**Keywords:** silver nanoparticles, shape, toxicity, *Hydra vulgaris*

## Abstract

Silver nanoparticles (AgNPs) are increasingly used in various consumer products, leading to their inadvertent release in aquatic ecosystems. The toxicity of AgNPs could be associated with the leaching of ionic Ag but also with the size, shape and surface properties. The purpose of this study was to test the null hypothesis that toxicity of AgNPs was independent of shape in the invertebrate *Hydra vulgaris*. The hydranths were exposed to increasing concentrations of ionic Ag and AgNPs of three different shapes (spherical, cubic and prismatic) with the same size and coating (polyvinylpyrrolidone). The data revealed that between 68% and 75% of total Ag remained in solution after the 96 h exposure period, while 85–90% of ionic Ag remained in solution. The 96 h lethal concentration (LC_50_) was lower with ionic (4 µg/L) and spherical AgNPs (56 µg/L), based on irreversible morphological changes such as loss of tentacles and body disintegration. Cubic and prismatic AgNPs were not toxic at a concentration of <100 µg/L. The sublethal toxicity was also determined at 96 h based on characteristic morphological changes (clubbed and/or shortened tentacles) and showed the following toxicity: ionic (2.6 µg/L), spherical (22 µg/L) and prismatic (32.5 µg/L) AgNPs. The nanocube was not toxic at this level. The data indicated that toxicity was shape-dependent where nanoparticles with a low aspect ratio in addition to high circularity and elongation properties were more toxic at both the lethal and sublethal levels. In conclusion, the shape of AgNPs could influence toxicity and warrants further research to better understand the mechanisms of action at play.

## 1. Introduction

In recent decades, the use of manufactured nanomaterials in a multitude of industrial and commercial applications has gained increasing interest. From basic to applied science, a sustained research effort has been observed in recent years, which has resulted in an increase in patent filings (Figure 1). The exponential growth of nanomaterial production is directly associated with unique properties conferred by nanomaterials’ physicochemical characteristics, including size, shape and surface reactivity [1]. Shape-related properties are now receiving more attention than ever before. By modulating their shape, nanoparticles can exhibit new specific properties while keeping the same core material [2]. Several economic sectors are trying to exploit this feature, ranging from the industrial field by increasing energy efficiency of thermal systems to health care services by improving the effectiveness of cancer treatments [3,4,5].

Among all engineered nanomaterials, silver-based ones predominate at a global scale owing to their presence in everyday consumer goods [6,7]. Known for their antimicrobial properties, silver nanoparticles (AgNPs) are found in textiles, personal care products and medical devices, among others [8,9]. They are also widely used in the field of electronics, where AgNPs are greatly exploited in conductive ink technology. In fact, these highly conductive printed elements can be found in various electronic devices such as touch screens, thin-film photovoltaic solar cells or sensors [10,11,12]. Despite the astonishing innovations created by their use, the darker side of the rising consumption of AgNPs is that it may also increase their potential release in aquatic ecosystems and thus have negative impacts on various organisms [13,14,15].

According to the current literature, adverse effects following an AgNP’s exposure can be associated with different mechanisms. Generally, their toxicity is related to cell surface interactions which can cause membrane damage and oxidative stress or a cellular uptake leading to mitochondria impairment, increased ROS generation and disruptive intracellular interactions [16,17,18]. The observed toxicity is also closely connected to the physicochemical parameters of the receiving environment, which can influence the stability of AgNPs. Indeed, numerous studies have shown that under certain environmental conditions, AgNPs’ transformation may occur, leading to nanoparticles’ aggregation, ion release or organic corona formation [13,19,20]. Over the years, a significant number of investigations have been conducted to better understand the effects of AgNPs’ properties on their toxic potential and related impacts on aquatic ecosystems. Although surface coating and size have been the most studied properties, some research groups have focused on understanding the shape-related effect. Despite the scarcity of studies, data were recently reported for fish, microcrustacean and algae showing differential toxicity between silver nanospheres, nanowires and nanoprisms [21,22,23,24,25]. However, with the data available, it is difficult to determine a clear trend in terms of shape-related toxicity because sometimes sizes are not in the same range or surface coatings are different. Moreover, a recent innovative biomarker development on freshwater mussels has also revealed the complexity surrounding the assessment of adverse effects related to nanoparticle shapes [26]. Finally, very few studies have explored the toxic effects of shapes other than the three mentioned above [26,27].

In view of this, further studies are needed to identify whether some silver shapes are of greater concern for the environment. To the best of our knowledge, no attention has been paid to this issue for the benthic aquatic invertebrate *Hydra*, which is a key organism in the trophic chain. Indeed, as a predator, *Hydra* contributes to maintaining the balance within planktonic communities [28,29]. In addition to its ecological importance, previous investigations have shown that *Hydra* is a particularly sensitive species for assessing the toxicity of nanoparticles and heavy metals [30,31]. The purpose of this study was to assess the toxicological effects of different forms of AgNPs (nanocube, nanosphere and nanoprism) on *H. vulgaris*, a well-recognized model for ecotoxicological studies [32,33,34,35].

## 2. Materials and Methods

### 2.1. Hydra vulgaris and Maintenance Conditions

The small tubular invertebrate *Hydra vulgaris* was cultured in crystallization bowels filled with 1 mM CaCl_2_ and 0.5 mM N-tris [hydroxymethyl] methyl 1–2-aminoethanesulfonic acid (TES buffer), pH 7.0 (*Hydra* medium) and maintained at room temperature under 16/8 h light–dark cycles, as described previously [36]. Animals were fed daily with newly hatched *Artemia salina*.

### 2.2. Nanoparticles and Reagents

The toxicity of three distinct shapes (spherical, cubic and prismatic) of AgNPs was studied on *H. vulgaris*. In an attempt to focus on size-related effects, the nanoparticles were in the same size range with the same coating. NanoXact^tm^ suspensions of silver nanospheres, nanocubes and triangular nanoprisms were purchased from nanoComposix, Inc (San Diego, CA, USA). Physicochemical characterization of each suspension was provided by the supplier in the certificate of analysis. According to transmission electron microscopy (TEM) analysis, particle sizes were in the same order of magnitude ranging from 74 nm to 86 nm in mean diameter. All selected AgNPs possessed a polyvinylpyrrolidone (PVP) surface coating, which is considered as a non-toxic polymer [37,38,39]. Silver nitrate (AgNO_3_) purchased from Millipore Sigma Canada Co. (Oakville, ON, Canada) was used as a comparator group for the ionic form. We examined the solubility of the various forms of nAg by Dynamic Light Scattering (DLS) and ion coupled mass spectrometry, using single particle mode as previously described [40] at 100 µg/L after 1 h and 96 h in the exposure media. The results showed that the Zeta potential did not change between the PVP-coated forms of nAg dissolved in the exposure media at 100 µg/L. Moreover, nominal Ag concentrations represented between 80% and 95% of the added concentration and dropped to 70–75% of the added values at 96 h.

### 2.3. Transmission Electron Microscopy (TEM) Image Analysis

TEM images provided by the manufacturer (Appendix A) were analyzed using the ImageJ software version 1.52n (National Institutes of Health, Bethesda, MD, USA) [41]. Area and perimeter measurements were carried out and then used to determine different shape descriptors (circularity, boxivity, elongation and aspect ratio). Calculation of circularity values (C) was performed according to the following formula where C = 4 π × area/(perimeter)^2^ [42]. Boxivity values were determined by dividing the area of the nanoparticle by the area of the bounding rectangle [43]. Elongation values were obtained by the ratio of the longest and shortest side of the bounding rectangle [44]. Aspect ratios were calculated for each nanoparticle by dividing their longest length (L) by their thickness (d) according to the following equation R = L/d where R is the resultant aspect ratio [45]. Surface area and surface-to-volume ratio were calculated using classical geometrical formulas (sphere, cube and prism).

### 2.4. Acute and Chronic Toxicity Assay

Toxicity assessments were conducted on *H. vulgaris* to determine lethal and sublethal effects of AgNPs with different shapes and their ionic comparator AgNO_3_ (dissolved Ag). Given that these organisms have a population doubling time of 3–4 days, exposure periods of 24 and 96 h were considered acute and chronic, respectively [46,47,48]. Exposures were carried out in a 12-well microplate with 3 individuals per well in triplicate (*n* = 9). Two-fold serial dilutions of the nanoparticles and dissolved Ag were prepared with *Hydra* medium, resulting in seven test concentrations ranging from 1.56 µg/L to 100 µg/L plus a control (*Hydra* medium). The acute and chronic toxicity assays were performed under static conditions at 20 °C without feeding for 24 h and 96 h, respectively. Observations on morphological integrity were scored for each *Hydra* based on the Wilby scale [49]. The sublethal endpoints are reversible and consisted of clubbed and/or shortened tentacles, while lethal irreversible signs consisted of the disappearance of tentacles (body appears as a tulip), condensation and disintegration. The collected data were used to calculate the lethal concentration (LC_50_) and sublethal effects (EC_50_) values.

### 2.5. Statistical Analysis

Lethal and sublethal ecotoxicity data were analyzed with CETIS software version 1.9.4.4. (Tidepool Scientific Software, McKinleyville, CA, USA). The 50% lethal concentration (LC_50_) and the sublethal effects (EC_50_) endpoints at 24 h (acute) and 96 h (chronic) of exposure were calculated using the Spearman–Karber method. Hierarchical clustering via the agglomeration approach was used to analyze similarities between silver nanoparticles of different geometries. Based on three shape nanodescriptors (elongation, boxivity and circularity), clusters were constructed using the complete linkage method and Euclidean distances as measure (Statistica version 7, Statsoft Inc., Tulsa, OK, USA). Tests for a monotonic trend in morphological changes were performed by using the Cochran–Armitage χ^2^ test for trends. Statistical significance of binominal proportion trends was set at *p* < 0.05. Data analysis was performed using GraphPad Prism, version 9 (GraphPad software, San Diego, CA, USA).

## 3. Results

### 3.1. Characterization of Nanoparticles Geometry

For each silver nanoparticle, TEM images were analyzed, enabling the calculation of six different nanodescriptors: surface area, surface-to-volume ratio, aspect ratio, circularity, boxivity and elongation (Table 1). Data revealed that although all AgNPs are in the same size range (74 nm–86 nm), silver nanoprism distinguishes itself from the others by a higher surface-to-volume ratio and aspect ratio, in addition to having lower value for the circularity (Table 2). On the other hand, boxivity is the descriptor that best characterizes silver nanocubes, knowing that a value equal to 1 represents a perfect square. Unsurprisingly, spherical nanoparticles obtained the highest value for circularity, which is a descriptor that considers area and perimeter.

Hierarchical clustering analysis based on shape descriptors was used for the nanoparticles grouping assessment (Figure 2). Among calculated nanodescriptors, elongation, boxivity and circularity were chosen because of their known ability to describe geometrical characteristics of nanoparticles [2,43,50]. Results demonstrate that the selected shape descriptors can effectively form well-defined clusters. The graphic also shows that nanoprisms greatly differ geometrically from the other two types of nanoparticles. As revealed by the hierarchical clustering analysis, cubic and spherical nanoparticles seem to share some similar characteristics even if they form distinct groups.

### 3.2. Silver Nanoparticles Toxicity of Different Shapes

Mortality was observed for *H. vulgaris* exposed to silver ions and nanosphere with LC_50_ values of 4.42 and 56.1 µg/L respectively after 96 h of exposure (Table 3). On the other hand, no mortality was found for the silver nanocube and nanoprism within the range of concentration tested. By looking more closely to data collected at 24 h of exposure, results confirmed that the mortality pattern observed for silver ions and nanospheres took place relatively early during the assay (Figure 3).

All sublethal ecotoxicity data related to morphological changes are shown in Table 3. Based on 96 h EC_50_ values, shape-dependent toxicity was found with AgNPs where nanosphere was the most toxic followed by AgNP prisms with EC_50_ values of 22.3 and 32.5 µg/L respectively. However, no significant toxicity was observed with the cubic AgNP. Lethal to sublethal effects ratio (LC_50_/EC_50_) values were determined to highlight shape-related effects in exposed *Hydra* (Table 3). The two most toxic forms of Ag (ionic and spheric) had decreased ratios compared to the less toxic ones, suggesting that sublethal effects occur at concentrations close to the lethal concentrations. For spherical AgNPs, this means that sublethal effects occurred at a concentration 2.5 times lower than the corresponding LC_50_.

The EC_50_ for the cubic AgNPs was more difficult to determine since it deviated from linearity. The Cochran–Armitage χ^2^ test for trend was performed on sublethal toxicity data to validate the presence of a monotonic concentration-response relationship (Figure 4). As expected, significant monotonic trends were found for nanosphere and nanoprism with *p* < 0.0001. Furthermore, results of the trend test confirmed that silver nanocube does not follow a monotonic dose–response curve (*p* < 0.34). In order to confirm a non-monotonic trend for the silver nanocube, a Fisher’s exact test was performed to highlight differences between proportions. Even though we observed a tendency, morphological changes were not statistically significant between the silver nanocube at 50 µg/L (highest maxima) and the control (*p* < 0.21).

## 4. Discussion

In this study, all the nanoparticles were in the same size range and with a PVP surface coating. This polymer is known to be a shape-directing agent in the synthesis of metallic nanoparticles and is widely used in the synthesis of cubic-shaped AgNPs [37,51]. In addition to its role in the production of various nanoparticle shapes, previous studies have demonstrated that PVP coating reduced the toxicity of AgNPs [52,53]. By limiting the variability of these physicochemical characteristics, it was shown that it was possible to reveal shape-related effects induced by nanoparticles [27,54].

The analysis of TEM images confirmed that all AgNPs tested were distinguishable based on shape descriptors, by forming three distinct groups. Hierarchical clustering analysis also revealed that prismatic particles occupy the most distant cluster in addition to their high aspect ratio and surface-to-volume ratio. Unlike the nanocube or nanosphere, the nanoprism falls in the category of a high-aspect-ratio nanoparticle (HARN), with a lateral dimension significantly larger than their thickness, i.e., possessing an aspect ratio of 3 or greater [55,56,57]. With a value of 8.6, the aspect ratio of nanoprism is in a similar range to that of nanorod, one of the well-recognized geometries in this category [45,58,59]. High aspect ratios are known to have an impact on the cellular uptake of nanoparticles by causing a delay if the internalization pathway requires a higher degree cell membrane deformation [60,61,62]. Moreover, this physicochemical property also drove their self-assembly; for example, when the aspect ratios are in the 5–15 range, nanoprisms tend to form ribbon structures instead of organized liquid crystals [63]. On the other hand, it was shown that surface-to-volume ratio correlates with the dissolution rates of the nanoparticle where an increase in surface-to-volume ratio contributes to a higher dissolution rate [38,64].

The toxicity evaluation of AgNPs in this study revealed a shape-related differential response with a 96 h LC_50_ value of 56.1 µg/L for the nanosphere and undefined values for both nanoprism and nanocube. Mortality values observed within the first 24 h of exposure (acute) were comparable to those obtained at the end of the 96 h (chronic). This finding is consistent with a recent study on the effects of metallic elements where toxicity was observed within 24 h of exposure [31]. Considering sublethal effects, the overall results indicated that AgNPs are less toxic than Ag^+^. However, from the perspective of a shape-related effect, nanosphere is the most toxic, followed by the nanoprism and nanocube. These results partly contradict previous studies reporting that nanoprism were slightly more toxic than nanosphere to *Danio rerio* embryos, *Daphnia magna* and *Chlorococcum infusionum* [22,23,24]. Nevertheless, our data are in line with those of Gorka et al. (2015), showing that nanocube was less toxic to zebrafish embryos than nanosphere. This lack of consistency with previous published data may be due to the size of the nanoparticles studied. Indeed, all these studies on PVP-AgNPs shape-related toxicity in aquatic toxicology have worked with nanoprisms whose size was less than 45 nm. By lowering their aspect ratio, nanoprisms in that size range have a greater cellular uptake and self-assembly processes come less into play, which could explain the higher toxicity observed in these studies [63,65]. Furthermore, the obtained sublethal results for the prismatic AgNPs used in the present study could also be associated with the formation of nanostructures. This phenomenon is more likely to be observed with an increase in nanoparticle aspect ratio, which can trigger their precipitation and decrease their toxic effect over time [13,63]. Leaching of toxic ions from silver nanoparticles is another mechanism that could contribute to this discrepancy. It has been shown that small size and high surface-to-volume ratio significantly increase the dissolution rate of AgNPs, leading to a higher release of Ag^+^ [64,66]. However, data in the literature are still contradictory regarding dissolution rates related to the geometry (Table 4) and highlight the complexity of environmental factors that can promote nanoparticle dissolution [67]. Moreover, patterns of dissolution can have a significant impact on toxicity outcomes. Cellular interaction and internalization could favor a well-defined spatiotemporal increase in toxic ions that could even exceed the concentration of the surrounding environment [68]. On the other hand, this process does not only increase their ionic counterpart but also generates a population of extremely small nanoparticles (1–30 nm) that possess unique quantum properties [69]. Considering all those variables under the angle of geometry, it would be possible to observe a shape-related toxicity that is higher for small nanoprisms (≤45 nm) than for larger ones, since they are prone to behave differently.

Sublethal data analysis revealed that the nanocube did not generate a monotonic dose–response curve as with the other AgNPs forms (spherical and prismatic) but rather adopted a response that was more consistent with a non-monotonic trend. In the current literature, this type of dose response is more often associated with receptor-mediated endocrine disruptors and hormones [70,71,72,73]. However, recent studies have shown that this type of response may also be related to other chemicals such as anti-diabetic drugs, alkaloids, pH indicators, metals and nanoparticles [69,74,75]. In order to better understand this complex dose response, Owen et al. (2014), demonstrated that colloidal aggregates were one of the major factors that could explain the non-monotonic response. Aggregation is known to impact the dissolution of NPs, with rates becoming slower as the surface-to-volume ratio of newly formed particles decreases [76]. Moreover, by increasing the concentration of NPs, formation of aggregates can occur, which is known to decrease the toxic response compared to lower doses containing a greater proportion of monodisperse NPs [69,77]. A recent study has demonstrated that self-assembly of NPs into organized nanostructures was related to their shape [78]. In fact, they showed that nanocubes were more prone to self-assembly than other forms due to their flat faces promoting stronger hydrophobic interactions. In light of all these findings, toxicity results for nanocubes could be explained by a self-assembly dynamic that is amplified with the increase in nanoparticle concentration. This would therefore lead to their marginal presence in the monomeric form, thus reducing their toxicity at higher concentration (100 µg/L).

## 5. Conclusions

In conclusion, the toxicity (LC_50_) of AgNP was shape-dependent in the following manner: ionic > spherical > cubic~prismatic. Sublethal results also revealed a differential toxicity related to the geometry of silver nanoparticles in the following manner: ionic > nanosphere > nanoprism > nanocube, suggesting that AgNPs with a low aspect ratio in addition to high circularity and elongation properties were associated with both lethal and sublethal toxicity in *Hydra*. This study highlighted the possibility that some forms of silver nanoparticles may be of greater concern. Further research is warranted to understand their toxicity at the molecular level.

## Figures and Tables

**Figure 1 nanomaterials-12-03107-f001:**
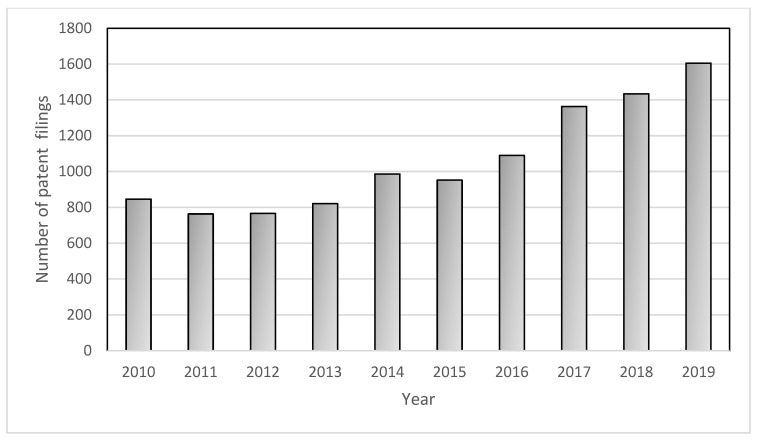
Number of patent filings listed in the WIPO Patentscope database that are related to the keywords nanomaterial and nanoparticle from 2010 to 2019 (“title” search by keywords).

**Figure 2 nanomaterials-12-03107-f002:**
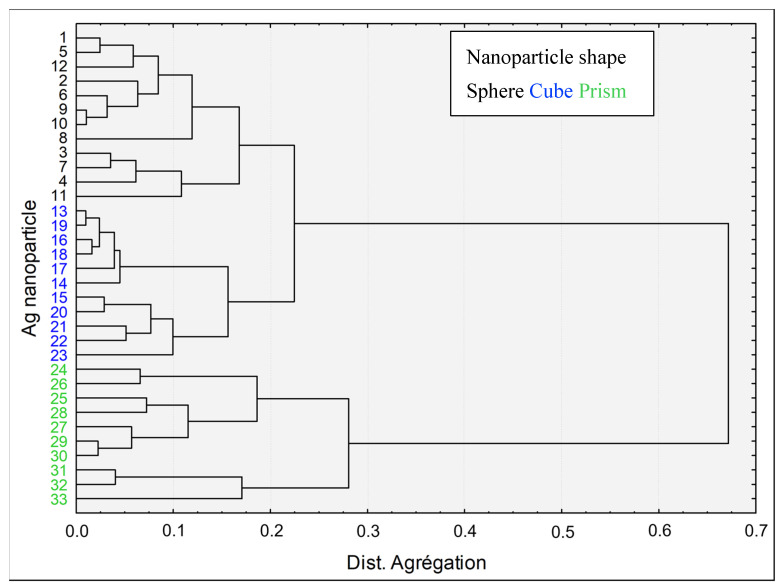
PVP-coated silver nanoparticles and hierarchical clustering based on shape nanodescriptors. The hierarchical clustering was built with three nanodescriptors (elongation, boxivity and circularity).

**Figure 3 nanomaterials-12-03107-f003:**
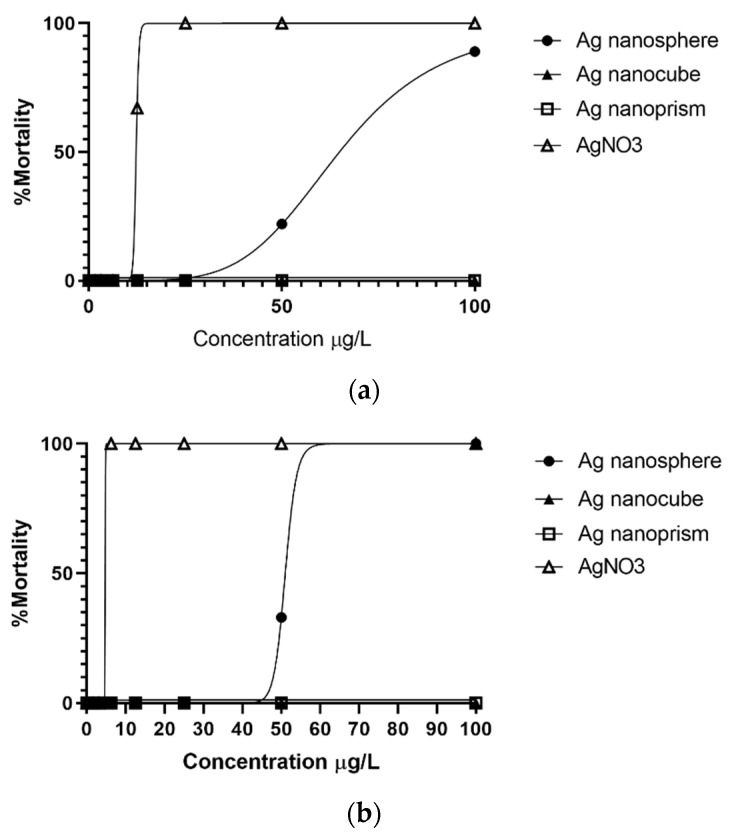
*Hydra vulgaris* percentage of mortality for PVP-coated silver nanoparticles and silver ions (AgNO_3_). Lethality dose–response curves are shown for time points at 24 h (**a**) and 96 h (**b**), which correspond to irreversible morphological stages (tulip and disintegrated stages) associated with a non-viable status.

**Figure 4 nanomaterials-12-03107-f004:**
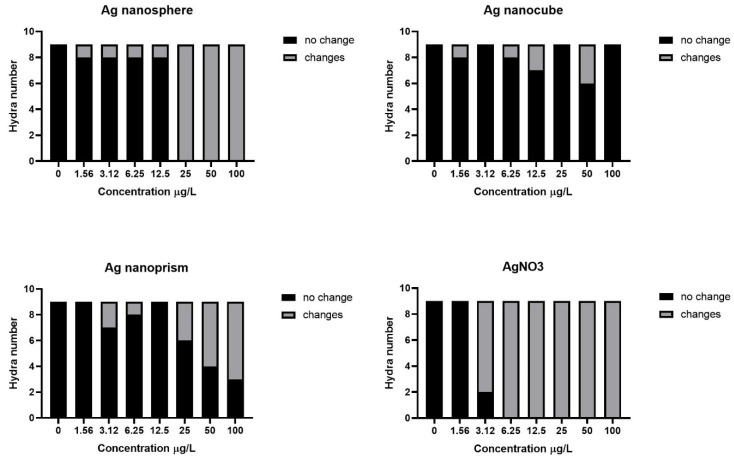
Stacked bar chart showing number of *Hydra* with and without sublethal morphological changes (clubbed and/or shortened tentacles) at 24 h of exposure to concentrations of AgNPs with different shapes. Cochran–Armitage χ2 test for trend: nanoprism (Z = 17.46; *p* < 0.0001); nanocube (Z = 0.9118; *p* < 0.34); nanosphere (Z = 43.51; *p* < 0.0001); AgNO_3_ (Z = 42.84; *p* < 0.0001).

**Table 1 nanomaterials-12-03107-t001:** Calculated nanodescriptors related to particle geometry.

Nanodescriptor	Definition	Formula Reference
Surface area (SA)	The total surface area of a particle specific to a particular geometry	Classical geometry formulas
Surface-to-volume ratio (SA:V)	Ratio of the surface area of a particle by its volume	Classical geometry formulas
Aspect ratio (AR)	Ratio of the greatest to the least dimension of a particle	[45]
Circularity	Degree of similarity between a particle and a circle where a value of 1 corresponds to a circular shape. Value between 0 and 1	[42]
Boxivity	Degree of similarity between a particle and a rectangle where a value of 1 corresponds to a rectangular shape. Value between 0 and 1	[43]
Elongation	Ratio of the largest side and the shortest side of the minimum bounding rectangle. This ratio expresses the degree of elongation of a particle whose square and circular shapes tend towards a value of 1. Value between 0 and 1.	[44]

**Table 2 nanomaterials-12-03107-t002:** PVP-coated silver nanoparticles, physicochemical properties and calculated nanodescriptors.

Nanodescriptor		Nanoparticle	
	Silver Nanosphere PVP-Coated	Silver Nanocube PVP-Coated	Silver Nanoprism PVP-Coated
**Size** **TEM (nm)**	76 ± 6	74 ± 7	86 ± 20
**Nominal Ag concentration in the exposure media ^1^**	1 h: 76 ± 6% 96 h: 71 ± 3%	1 h: 81 ± 6% 96 h: 69 ± 3%	1 h: 77 ± 5% 96 h: 72 ± 4%
**Zeta potential** **(mV)**	−41	−38	ND
**Surface area** **(nm^2^)**	16,742	32,856	8985
**Surface-to-volume ratio** **(nm^2^/nm^3^)**	0.0822	0.0811	0.2806
**Aspect ratio**	1	1	8.6
**Circularity**	0.776 ± 0.020	0.680 ± 0.051	0.363 ± 0.056
**Boxivity**	0.764 ± 0.026	0.901 ± 0.018	0.602 ± 0.055
**Elongation**	0.983 ± 0.069	0.955 ± 0.034	0.932 ± 0.025

^1^ The levels of total Ag were determined in the exposure media for the 5 and 50 ug/L nAg after 1 and 96 h dissolution. Ionic Ag remained in solution at 80–90% of the added concentration (5 ug/L). ND: not determined.

**Table 3 nanomaterials-12-03107-t003:** Lethal (LC_50_) and sublethal (EC_50_) responses in µg/L with 95% confidence intervals (CI) for PVP-coated silver nanoparticles and silver ions (AgNO_3_).

	Silver Form			
	Ag^+^	Silver Nanosphere PVP-Coated	Silver Nanocube PVP-Coated	Silver Nanoprism PVP-Coated
**24 h EC_50_ (CI)**	2.58 (2.13–3.13)	16.93 (15.44–18.56)	ND	51.89 (33.24–101.2)
**24 h LC_50_ (CI)**	11.14 (8.96–13.85)	65.52 (48.47–89.67)	ND	>100
**96 h EC_50_ (CI)**	2.59 (2.20–3.05)	22.3 (17.9–27.7)	ND	32.5 (22.6–48.5)
**96 h LC_50_ (CI)**	4.42 (3.12–6.25)	56.1 (45.1–69.8)	ND	>100
**LC_50_/EC_50_** ratio 24 h	4.3	3.8	ND	>1.9
**LC_50_/EC_50_** ratio 96 h	1.7	2.5	ND	>3.1

ND: not determined.

**Table 4 nanomaterials-12-03107-t004:** Reported AgNPs dissolution rate differences between nanocube, nanosphere and nanoprism.

AgNPs Dissolution Rate	Surface Coating	Medium	Reference
Nanophere > Nanocube	PVP/PVP	ultra-pure water (pH 4.8)	[64]
Nanophere = Nanoprism	PVP/PVP	Holtfreter’s medium (pH 6.5–7)	[23]
Nanophere > Nanoprism	PVP/PVP	acetate buffer (pH 4)	[60]

## Data Availability

Data could forwarded upon demand.

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
