# Peer review of "Shape-Dependent Toxicity of Silver Nanoparticles on Freshwater Cnidarians"

_nanomaterials, 2022, doi:10.3390/nano12183107_

Round 1
Reviewer 1 Report
The development of nanomaterials opens new trends in modern toxicology and ecology related to the safety of nanomaterials. The presented paper describes the impact of the shape of the silver nanoparticles on their toxicity against freshwater cnidarians. The paper can be accepted for publication after major revision. The following issues should be clarified.
Why was Hydra vulgaris chosen as the modeling organism? Is it related to the protection of biodiversity? Please add appropriate discussion.
Discussion about mechanisms of toxicity should be provided. Is it an impact of the silver ions or contact of the nanoparticles with the organism's surface or penetration of the nanoparticles into Hydra vulgaris. Why silver nanocube does not follow a monotonic dose-response curve? Please add appropriate discussion.
What about the impact of polyvinylpyrrolidone on the toxicity of silver nanopartilces?
The last years intensively developed the trend where silver nanoparticles are immobilized on carriers to reduce their toxicity. Please cite appropriate references:
https://doi.org/10.1016/j.colsurfa.2022.128525
https://doi.org/10.1039/C9RA10874B
Author Response
Reviewer 1
The development of nanomaterials opens new trends in modern toxicology and ecology related to the safety of nanomaterials. The presented paper describes the impact of the shape of the silver nanoparticles on their toxicity against freshwater cnidarians. The paper can be accepted for publication after major revision. The following issues should be clarified.
Why was Hydra vulgaris chosen as the modeling organism? Is it related to the protection of biodiversity? Please add appropriate discussion.
Hydra was included as sensitive and another species of the invertebrate group (It is a large taxonomic group and more members should be included in risk assessment). This was added in the revision; please see lines 71-74.
Discussion about mechanisms of toxicity should be provided. Is it an impact of the silver ions or contact of the nanoparticles with the organism's surface or penetration of the nanoparticles into Hydra vulgaris. Why silver nanocube does not follow a monotonic dose-response curve? Please add appropriate discussion.
The section explaining the toxicity of nanoprisms (lines 222-243) has been rewritten to better describe the mechanisms involved. Please see line, 224-231 and 233-234. Cellular uptake, self-assembly processes and leaching of silver ions are discussed in the text.
In the discussion, explanations regarding the toxicity of silver nanocubes are found in the lines 245-263. Toxicity response of silver nanocube can be explained by self-assembly processes that would dominate at higher concentrations. This would lead to marginal presence of nanocubes in the monomeric form thus modifying the toxicity profile. The section explaining this phenomenon was modified to better defined this concept. Please see line 257-263
What about the impact of polyvinylpyrrolidone on the toxicity of silver nanoparticles?
The studies we referred to show that PVP alone is non-toxic. This was added in the revision, please see lines 94-96.
The use of PVP nanosilver in this study was also to give us the opportunity to explore the effect of different shapes, as PVP is mainly use in the synthesis of nanocube and other shapes. This was added in the revision, please see lines 191-193.
Other studies have shown that PVP was able the reduced the toxicity of AgNPs. This was added in the revision, please see lines 193-195
The last years intensively developed the trend where silver nanoparticles are immobilized on carriers to reduce their toxicity. Please cite appropriate references:
These studies are very interesting and are more in line with our future research projects, I prefer to save your references for this one.
https://can01.safelinks.protection.outlook.com/?url=https%3A%2F%2Fdoi.org%2F10.1016%2Fj.colsurfa.2022.128525&data=05%7C01%7Cjoelle.auclair%40ec.gc.ca%7Cb7b52d96242745d6514708da7fa3c9d5%7C740c5fd36e8b41769cc9454dbe4e62c4%7C0%7C0%7C637962639548708592%7CUnknown%7CTWFpbGZsb3d8eyJWIjoiMC4wLjAwMDAiLCJQIjoiV2luMzIiLCJBTiI6Ik1haWwiLCJXVCI6Mn0%3D%7C3000%7C%7C%7C&sdata=l%2FnHyfkRPE9JhDWxKCwah36mKn0aqAGquooNxvBqevc%3D&reserved=0
https://can01.safelinks.protection.outlook.com/?url=https%3A%2F%2Fdoi.org%2F10.1039%2FC9RA10874B&data=05%7C01%7Cjoelle.auclair%40ec.gc.ca%7Cb7b52d96242745d6514708da7fa3c9d5%7C740c5fd36e8b41769cc9454dbe4e62c4%7C0%7C0%7C637962639548708592%7CUnknown%7CTWFpbGZsb3d8eyJWIjoiMC4wLjAwMDAiLCJQIjoiV2luMzIiLCJBTiI6Ik1haWwiLCJXVCI6Mn0%3D%7C3000%7C%7C%7C&sdata=uPiTqSRJ1cH9hcxC8ONhDr3HlB4J7eWPu%2FBJscMRDkw%3D&reserved=0

Reviewer 2 Report
The work presented herein sounds interesting, dealing with the “Shape-dependent toxicity of silver nanoparticles on freshwater cnidarians”.
Before acceptance, there are some issues that should be addressed.
1. Line 137, please put the word Results in line with the rest of the text.
2. The shape of Table 1 differs to that of the rest. For homogeneity, please change it appropriately.
3. In Table 2, all numbers should be written in a same way not with comma.
4 . In Table 3, please specify the meaning of numbers in ( ).
5. Line 210, please change obtain to obtained .
6. Through the text, please change Ag+ to Ag+ that is scientifically correct.
7. Line 231, please change well-define to well-defined.
8. References should be appropriately written, which is not the case. In some of them, complete page numbering is missing, name of the journal is not correctly written. Authors should check them thoroughly.
9. In Fig. 4. The authors are speaking about dose-response curves. However, these are histograms. Please specify.
To summarize
The article merits publication. I suggest minor revision. Provided that the authors take my comments into consideration I could reconsider my opinion.
Author Response
Reviewer 2
Before acceptance, there are some issues that should be addressed.
- Line 137, please put the word Results in line with the rest of the text.
Corrections were made, please see line 139
- The shape of Table 1 differs to that of the rest. For homogeneity, please change it appropriately.
Shape of table 1 was changed in the revision. Please see the new version in the manuscript.
- In Table 2, all numbers should be written in a same way not with comma.
Corrections were made, please see Table 2
4 . In Table 3, please specify the meaning of numbers in ( ).
The information was added in the revision. Please see the new version in the manuscript.
- Line 210, please change obtain to obtained.
Corrections made, please see line 215.
- Through the text, please change Ag+ to Ag+ that is scientifically correct.
Corrections made, please see line 217 and table 3.
- Line 231, please change well-define to well-defined.
Corrections made, please see line 237.
- References should be appropriately written, which is not the case. In some of them, complete page numbering is missing, name of the journal is not correctly written. Authors should check them thoroughly.
Corrections were made for the reference section
- In Fig. 4. The authors are speaking about dose-response curves. However, these are histograms. Please specify.
The title was changed. Please see figure 4 in the manuscript.
Reviewer 3 Report
The present work raises an interesting question. However, authors don't make any attempt to explain the observed result, which diminishes the quality of the article.
While a great statistical analysis was performed, no conclusions are really made based on this analysis. One of the possible explanation for the observed toxicological effect is that particles of different shape have different crystal planes developed and exposed. The mortality might also depend on the ability of particled of certain shape to release ions into the medium. Existing literature on the topic must be analyzed more thoroughly.
The study also lacks a control experiment with just PVP added to the Hydra. While all the purchsed particles have the same coating, it is known that a coating might affect the result, so this control experiment is necessary to check is PVP has any share in the mortality rate (or decrease thereof). Otherwise, literature data must be provided to prove its non-toxicity.
Below are some more small improvements that must be made to the text:
1. The references must be given as numbers in square brackets, as per the journal's standard.
2. The manuscript must be checked for typos. For example, typos are present 13, 65 and 137, but those are certainly not the only ones.
3. Fig.1 does not seem relevant. The trend does not really seem to be exponential. The exact search keywords are not given.
4. In line 78 it would be fitting to describe, what other model organisms were used for similar studies. Depending on the type of organism, the effect of inorganic nanoparticles might be different.
5. In line 84 TES is mentioned for the first time and the meaning of this abbreviation should be given.
6. The TEM images and the results of data mentioned in line 184 must be given in supplementary section.
7. Fig.2 requires more discussion, as currently it is not clear how it confirms the statement in line 163.
Author Response
Reviewer 3
The present work raises an interesting question. However, authors don't make any attempt to explain the observed result, which diminishes the quality of the article.
While a great statistical analysis was performed, no conclusions are really made based on this analysis. One of the possible explanation for the observed toxicological effect is that particles of different shape have different crystal planes developed and exposed. The mortality might also depend on the ability of particled of certain shape to release ions into the medium. Existing literature on the topic must be analyzed more thoroughly.
The section explaining the toxicity of nanoprisms ( lines 222-243) has been reworked to better describe the mechanisms involved. Please see line, 224-231 and 233-234. Cellular uptake, self-assembly processes and leaching of silver ions are discussed in the text.
In the discussion, explanations regarding the toxicity of silver nanocubes are found in the lines 245-263. Toxicity response of silver nanocube can be explained by self-assembly processes that would dominate at higher concentrations. This would lead to marginal presence of nanocubes in the monomeric form thus modifying the toxicity profile. The section explaining this phenomenon was modified to better defined this concept. Please see line 257-263
The study also lacks a control experiment with just PVP added to the Hydra. While all the purchsed particles have the same coating, it is known that a coating might affect the result, so this control experiment is necessary to check is PVP has any share in the mortality rate (or decrease thereof). Otherwise, literature data must be provided to prove its non-toxicity.
The studies we referred to show that PVP alone is non-toxic. This was added in the revision, please see lines 94-96.
Below are some more small improvements that must be made to the text:
The references must be given as numbers in square brackets, as per the journal's standard.
Corrections were made
The manuscript must be checked for typos. For example, typos are present 13, 65 and 137, but those are certainly not the only ones.
All the manuscript was rechecked for typos.
Fig.1 does not seem relevant. The trend does not really seem to be exponential. The exact search keywords are not given.
Figure 1. Presents our results search from the WIPO database, and support the fact an increase in patent filings was observed in recent years, please see line 30. Keywords used for the search were added in the revision, please see the description of figure 1.
The exponential trend state in line 31 is an additional information that refers to the production of nanomaterials and not to patent fillings.
In line 78 it would be fitting to describe, what other model organisms were used for similar studies. Depending on the type of organism, the effect of inorganic nanoparticles might be different.
In lines 60-64, you can find information about the different studies that have also explored the shape-dependent toxicity of AgNPs on different aquatic organisms (fish- zebrafish and Oryzias latipes, microcrustacean - Daphnia magna and Daphnia galeata, algae- Chlorococcum infusionum and Raphidocelis subcapitata).
In line 84 TES is mentioned for the first time and the meaning of this abbreviation should be given.
TES refers to N-tris [hydroxymethyl] methyl 1-2- aminoethanesulfonic acid. This was added in the revision, please see line 83-84.
The TEM images and the results of data mentioned in line 184 must be given in supplementary section.
TEM images from the manufacturer were added in the revision. Please see the supplementary section.
P value for the Fisher's exact test and additional information was added in the revision. Please see line 186-188.
- Fig.2 requires more discussion, as currently it is not clear how it confirms the statement in line 163.
Regarding the following statement:
‘’results confirmed that the mortality pattern observed for silver ions and nanospheres take place relatively early during the assay (figure 2).’’
This statement refers to figure 3, corrections were made, it should be more clear now. Please see line 163-166. All figures mentioned in the manuscript have been rechecked.
Round 2
Reviewer 1 Report
After revision, the paper can be accepted for publication in its present form.
Author Response
After revision, the paper can be accepted for publication in its present form.
Thank you, F Gagné
Reviewer 3 Report
Dear authors!
Thank you for the attention to my remarks and the changes that you made to the text. The text is much easier to read now, in my opinion, and the manuscript can be published in present form.
Author Response
Thank you for the attention to my remarks and the changes that you made to the text. The text is much easier to read now, in my opinion, and the manuscript can be published in present form.
Thank you, F Gagné